# Data from Multiple Portable XRF Units and Their Significance for Ancient Glass Studies

**DOI:** 10.3390/molecules27186068

**Published:** 2022-09-17

**Authors:** Oleh Yatsuk, Marco Ferretti, Astrik Gorghinian, Giacomo Fiocco, Marco Malagodi, Angelo Agostino, Monica Gulmini

**Affiliations:** 1Department of Chemistry, University of Turin, Via Giuria 7, 10125 Torino, Italy; 2Italian National Research Council, Institute of Heritage Sciences, A.d.R. RM1, Via Salaria km 29.300, 00015 Montelibretti, Italy; 3INFN—LNF (National Laboratory of Frascati), Via Enrico Fermi 40, 00044 Frascati, Italy; 4Arvedi Laboratory of Non-Invasive Diagnostics, CISRiC, University of Pavia, Via Bell’Aspa 3, 26100 Cremona, Italy; 5Department of Musicology and Cultural Heritage, University of Pavia, Corso Garibaldi 178, 26100 Cremona, Italy

**Keywords:** portable-XRF, ancient glass, reproducibility, composition

## Abstract

X-ray fluorescence spectroscopy is a non-destructive technique employed for elemental analysis of a wide range of materials. Its advantages are especially valued in archaeometry, where portable instruments are available. Considering ancient glass, such instruments allow for the detection of some major, minor, and trace elements linked to the deliberate addition of specific components or to impurities in the raw materials of the glass batch. Besides some undoubted advantages, portable XRF (p-XRF) has some limitations that are addressed in this study. The performance assessment of four different p-XRF units and the reconciling of their output were conducted. The results show the limitations in cross-referencing the data obtained from each unit and suggest procedures to overcome the issues. The p-XRF units were tested on the set of Corning reference glasses and on a small set of archaeological glasses with known composition. The compatibility of the output was assessed using multivariate statistical tools. Such a workflow allows us to consider data from multiple sources in the same frame of reference.

## 1. Introduction

XRF analyses of archaeological materials conducted with portable instruments (p-XRF) are increasingly popular within the archaeometry community and have a history of development that spans decades [1,2,3,4,5]. Portable equipment has also been applied to investigate vitreous materials and ancient glass in particular [6,7,8,9]. The most relevant advantage of p-XRF is the non-invasiveness of the analytical approach, which is much appreciated by the institutions entrusted with the safekeeping of objects of historical value. Generally, no sample preparation is required unless the object’s surface is clearly contaminated, altered, or coated by conservation treatments. The time required for the analysis may vary. Anyhow it is possible to collect data from a large number of analytical spots within a single session, with qualitative information available in, practically, real-time. Producers of commercially available p-XRF indicate the limits of detection (LOD) as low as several ppm, though the actual LODs depend on the matrix composition [10].

Unfortunately, p-XRF has several drawbacks for the investigation of ancient glass. The most evident is a significant limitation in providing accurate information for “light” elements (Z < 14), which may constitute a large part of the total mass fraction in the glass matrix. This may inhibit a characterization of the fluxing agents used in glass making, which is one of the first research questions to be faced when studying archaeological glass. To overcome the issue, the p-XRF units can be equipped with helium-flow setups or with small chambers for vacuum analyses to reduce the absorption of X-rays by Ar from the atmosphere and enhance the possibility of detecting the weak XRF signals from Na and Mg [11,12,13].

The above-mentioned issues combine with those inherent to the general XRF analytical approach. Characteristic fluorescence lines of different elements may overlap, with typical examples relevant for glass studies being the interference of Fe Kβ and Co Kα, or that of Pb Lα and As Kα. These phenomena limit the accuracy of the quantification and may prevent elements from being detected if they are present in minor quantities [14]. Moreover, the measurement capability of XRF devices must undergo evaluation for every matrix that is expected to be studied.

Adjustment of the acquisition parameters can improve the quality of the data and might expand the number of applications for p-XRF as well as mark their limits [15,16,17].

Under the p-XRF label, we can find many types of spectrometers, with a variety of features [3]. Moreover, among the numerous examples of the application of p-XRF in archaeological glass studies, several approaches have been suggested to handle the data. The selection of the most suitable approach for data handling depends on the archaeological question to be answered [18], and the quality of data is often connected to the user’s level of expertise in the method [19]. Although an exhaustive review of the literature dealing with p-XRF applications for characterizing ancient glass is beyond the scope of the present paper, we can highlight, within the most recent literature, the approaches that are still in use.

Some users explore the spectra in a purely qualitative way: as long as energies are associated with elements and the position and height of the peaks correspond to the respective element’s presence/concentration, the comparison of the relative heights of the peaks provides information on the qualitative composition of the examined samples [20], though recently this approach has been mostly reserved for XRF imaging of glass objects [21,22].

The so-called “semi-quantitative” approach is based on different workflows. This approach does not allow direct comparison of the results with external data. Therefore, results obtained with different instruments cannot be directly compared. Nevertheless, it enables the possibility of distinguishing groups of artifacts within the obtained dataset. The basic assumption in this approach is that the collected data are similar but may be biased from an accepted nominal value. Using the definitions given in ISO 5725-1 [23], we shall say that the precision of the data is good, but trueness can be poor. Often, in this framework, p-XRF is used as a tool to optimize sampling for further analysis [24]. In such a mode, the concentration values are not a prerequisite for data interpretation and can be used in the same way as counts. Net area counts of peaks can be normalized to the intensity of the anode Compton or Kα peak. This step eliminates the potential differences in distances, angles, and matrix effects and allows for the comparison of acquisitions made with the same XRF unit without mass fraction quantification [25]. Researchers may operate on both counts per second (cps), weight percentages, or ppm concentrations [26].

Some operators use face values of factory calibration provided by the p-XRF unit [27]. They can convert them to oxides and normalize them to 100% [28,29] or use SiO_2_ values to compensate for the difference in total composition to 100% [30]. Regarding the light elements, such as Na and Mg, some researchers use average values for studied glass types [31] or estimate their content through the “dark matrix” calculated by the p-XRF processor [32]. Again, the approach yields equipment-dependent data with no comparative potential.

Accurate concentrations can be obtained when standard or reference materials are used to calibrate the spectra. This approach yields the highest quality results. The output of the process should in principle be comparable with results obtained from other p-XRF units or other types of analytical equipment. Some producers of p-XRF units prepare built-in calibrations that can be used when analysing archaeological glass or prepare them by request, using the Fundamental Parameters (FP) method or the Compton normalization [31,33]. The FP approach is one of the several methods for the quantification of XRF data and enables the accurate estimation of major, minor, and common trace elements in the glass matrix [4,5,28,29,33,34,35].

In general, the above-mentioned examples demonstrate that p-XRF is widely used in the analysis of archaeological glass, although most of the matrix remains undetected. Despite extensive applications to ancient glass, many researchers rely on other methods for quantitative compositional information, reducing p-XRF to a tool for preliminary analysis and using the p-XRF values in a strictly qualitative or semi-quantitative way.

Recently, the need for implementing integrated approaches between chemistry and archaeology has significantly grown within the entities that are responsible for studying and protecting cultural heritage. As a consequence, many relevant archaeological museums and research institutions own analytical equipment, with p-XRF units being among the most popular, as there are many research questions or purposes that can be answered or fulfilled by this technique. Consequently, an extended archaeometric investigation may face the issue of having different p-XRF units available for analysing multitudes of archaeological objects in different institutions. As the collected data needs to be compared to produce a new knowledge on the investigated collections, we must exclude any bias introduced by the use of different equipment.

There have been several papers reporting a comparison of the results obtained with different p-XRF units. They are mostly related to environmental samples and metals, whereas, to the best of the authors’ knowledge, none deals specifically with archaeological glass. Some of these papers highlight specific issues that cannot be overlooked while assessing the reproducibility of data from different instruments. In particular, it has been demonstrated how the data treatment can influence the final output [36] and that the low Z element values (Al and Si) have extremely low reproducibility [37]. The use of quality control references enhances the possibility to discriminate among samples of similar composition [38] and the significant influence of sample conditions during the analysis has been highlighted [39]. The CHARMed PyMCA protocol was an attempt to standardize the practice of XRF analyses of copper alloys by using the Fundamental Parameters (FP) approach combined with the use of calibration standards. This unified procedure improved the inter-laboratory reproducibility of the results and provided a framework for the validation of such results [40,41].

The present work aims at highlighting the main limitations in reproducibility of p-XRF analyses performed with units of different configurations (anode material, acquisition geometry, spot size, energy resolution, and power-voltage parameters), thus shedding light on the actual possibility to face the main archaeological questions for glass with a p-XRF approach if different instruments are employed. The study is preliminary to an analytical campaign that will be conducted in situ on large collections of Iron Age glasses held in several archaeological museums in Italy. The focus is set on quantification using FP combined with calibration using the reference glass approach, which would allow results to be exploited throughout a specific research project and beyond.

## 2. Materials and Methods

### 2.1. Reference Glasses

Certified Reference Materials (CRM) for glass that are offered on the market provide a wide selection of elements with certified or informational concentration values. Yet, the resemblance of the compositional range of ancient glasses with these (modern) reference materials remains poor. Therefore, for this study, we selected a set of three reference glasses provided by the Corning Museum of Glass (CMOG). The CMOG reference glasses were developed to represent different compositional groups of historic glass, and they feature the elements that play a role in glass (de)coloring and opacification, as well as some common impurities that enter the glass batch with the silica source or with other glass components. Each glass piece is named with letters for different glass types. CMOG A and CMOG B are soda-lime-silica glasses that represent typical soda plant ash (A) and natural evaporites (B) varieties of Mediterranean production glasses from the Bronze Age to the Medieval times [42]. CMOG D is a potash-lime-silica composition and was prepared to reflect the compositions of some medieval and 17th–19th century European glasses [43], though potassium-rich glasses existed earlier [44,45].

The compositions of CMOG glasses have been established through an inter-laboratory study, and since the 1970s, the glasses have been tested and their concentration revised according to the new results [46,47]. The compositions relevant for this study are reported in Table 1.

In addition to the CMOG reference glasses, five pieces of archaeological glass (henceforth “archaeological references” or ARCH) were also included in the investigated glass set and their composition is reported in Table 1. These samples originated from the Veh-Ardashir (Iraq) and Crypta Balbi (Italy) assemblages. They are indicated with the codes VA and CB, respectively, followed by a number that identifies the specific sample. Their compositions were determined by inductively coupled plasma-optical or mass spectrometry (ICP-OES or ICP-MS) [48,49,50,51].

In this work, VA08 and VA27 were analysed as chunks of glass, while VA70 and the CB samples were analysed as cross-sections. This allows us to evaluate the role of the shape of the surface on the quantification. Successful quantification of irregularly shaped samples will substantiate the quantification on future analyses to be performed non-invasively on archaeological glasses. The CMOG reference glasses underwent polishing to ensure the correct geometry of the acquisitions. Grinding paper of 500, 1200, and 2400 grit was used on flat pieces (about 1 cm^2^ area and 2–3 mm thick). In addition, they were polished using diamond pastes of 6 and 1 µm grain size.

### 2.2. XRF Units and Settings

Four p-XRF units were included in this study:XGLab ELIO (hereafter ELIO). The commercial unit is produced by XGLab S.R.L. (Milan, Italy);Unisantis XMF-104 (hereafter Unisantis). The commercial unit is equipped with polycapillary optics and produced by Unistantis (Geneva, Switzerland). Regarding the size and configuration of the instrument, this is the only one with limited sample size analysis due to the dimensions of the sample chamber and can therefore be included in the category of transportable X-ray spectrometers [52];Frankie (hereafter Frankie). An ad hoc unit with policapillary optics, developed by the Italian National Institute of Nuclear Physics-National Laboratories of Frascati ((INFN-LNF) Italy);NITON XL3T-900 GOLDD (hereafter NITON). The commercial unit is by Thermo Fisher Scientific (Waltham, MA, USA). This instrument acquires a series of spectra in different energy ranges to enhance the signal in specific parts of the spectrum.

The most relevant technical features of the equipment and the acquisition parameters used in this study are reported in Table 2. Obviously, the four p-XRF spectrometers have very different configurations that are reflected in the resulting spectra. Therefore, each unit required different parameter settings for the acquisition to achieve optimal results.

### 2.3. Acquisition and Processing

On each CMOG reference glass, 10 spectra (ELIO and NITON units) or 3 spectra (Unisantis and Frankie units) on different spots were acquired. For the archaeological references, 3 measures on different spots were acquired for each p-XRF unit, except for Frankie, for which only one acquisition was performed on each archaeological reference.

The obtained spectra were converted to .spc, .spe, or .mca before fitting with the PyMCA software [53]. The configuration of each XRF unit was used to prepare an instrument-specific FP configuration file which provided mass fraction values. The CMOG A reference glass spectrum was the one used to improve quantification by adjusting the incident primary spectrum profile to calibrate the FP composition with the reference one. Matching was done for 15 elements: K, Ca, Ti, Mn, Fe, Co, Ni, Cu, Zn, Sr, Zr, Sn, Sb, Ba, and Pb. For quantification, K series peaks were used for all except Pb and Ba, which were quantified using L series peaks. Elements such as Na, Mg, Si, and Al were not included in the fitting and were not considered in this study except for the FP model building, where nominal compositions of CMOG glasses including these elements were used. In the batch fitting mode, all the spectra were processed, and mass fraction values of the elements were calculated and systematized. 

The Limit of Quantification (LOQ)—the lowest amount of analyte that can be quantified with a given uncertainty—for each element’s concentration was estimated by using the value of the gross area B under the relative peak (Kα for most of the elements except Ba and Pb, where L series peaks were used). The LOQ was set to be equal to B + 10σB. This corresponds to an uncertainty of ±30% in the measured value at the 99% confidence level [54]. This value was found by empirical matching of concentrations to areas of the peaks as described in the PyMCA tutorial [55]. This procedure was performed on CMOG A, B, and D spectra, and the maximum value of LOQ (individual for each element of interest) for these glasses was used as the general LOQ for all the samples. All the values that were below the LOQ estimation were discarded.

The relative standard deviation (RSD) of tri- and decaplicates was used to calculate measurement precision. To check the accuracy, the values were converted to elemental weight % and compared to the nominal concentration (Table 1). In order to improve the accuracy, linear regression correction was applied. Accuracy is reported in Section 3.3 as the double mean relative error for each element (2σ values). Another way of correcting the data is the normalization of the acquired values by the mean ratio of nominal to acquired values of CMOG glasses. This method may be useful when the number of data points for building the regression curve is less than 4. Data corrected in this way were taken as semi-qualitative because of the impossibility of reporting the uncertainties. General reproducibility of the approach was judged by the results of Principal Component Analysis and Hierarchical Cluster Analyses, where the condition of success was the grouping together of data on the same sample obtained with different devices, thus excluding equipment bias as a source of variability for the sample set.

## 3. Results and Discussion

### 3.1. Estimation of Limits of Quantification

In order to ensure that all the data points are a real representation of the concentration in each of the references, the Limits of Quantification (LOQs) were calculated. The automated fitting of the spectra assigned values to the selected elements even when there were no pronounced peaks in the spectra. Therefore, LOQs are needed to filter out the data that were not significant for the quantification process. According to this procedure, the data that were equal or below the LOQ were removed before any precision and accuracy checks.

There are several ways of establishing LOQs [54,56,57]. As outlined above, for practical reasons, a 10σ threshold of the background deviation was selected to represent the LOQs in the present study. Figure 1 reports the calculated LOQs as the elemental concentration for the element of interest. They are the maximum values of the LOQ estimation for CMOG A, B, and D reference glasses for each of the p-XRF units. The element concentrations in the CMOG glasses are included in the graph to simplify the comparison.

LOQs in XRF analyses usually depend on the matrix composition and the element’s atomic number. It is lower for elements with Z ranging between 25 and 40, and higher for lighter ones. The features of the source of radiation are also important to determine LOQ, as source peaks due to Compton and Rayleigh scattering might interfere with the characteristic energy peaks of the analysed elements, raising the LOQs. The same effect is observed when the characteristic energies of the elements in the analysed sample overlap in the detected spectrum.

Figure 1 demonstrates that the estimated LOQs are all in accordance with the expected tendencies. It can be noticed that most of the elements are either under or within the range of concentration of CMOG glasses. LOQ estimation for the NITON unit is above the concentration range of CMOG glasses for Ni and Ba. These elements were detected during the fitting, although the intensity of their signals was below the LOQ. The elements with LOQs within the range of CMOG glasses concentration were detected in at least one of the reference materials. Discarding the data points below LOQ makes the dataset rather small for some elements (Figure 2).

Unfortunately, for many minor elements, the number of data is equal to or less than five. Therefore, calculated uncertainty may not reflect the true value for these elements. Larger datasets are available for K, Ca, Mn, Fe, Cu, Sr, and Pb. Some elements (Co, Zn, Zr, Sn, and Ba) may have only one data point considered by a single instrument (there might be more data for another instrument). Data on fewer represented elements will reflect poorer precision, and this will be considered in the further discussion of the data. Furthermore, because of their heteroscedastic nature, quantitative data near the LOQ should be handled with caution [40].

### 3.2. Precision of the Measurements

The variability among the repeated measures was estimated as the Relative Standard Deviation (RSD). Mean RSD values per unit of equipment, together with their respective maximum values, are shown in Figure 3.

It can be noticed that, generally, the relative dispersion of the data is less than 20% for most of the elements across all the XRF units. The polished cross-sections of CMOG glasses do not differ significantly from the sections and the chunks of archaeological references. The only exceptions are some of the Unisantis acquisitions (VA27 was discarded from consideration altogether because of extremely high RSD values, VA70 has one of the highest RSDs for this instrument). One of the reasons for high RSD values might be the proximity of the measured values to the respective LOQ (as in the case of ELIO CB65 Sn (68%RSD) and CMOG B Co (36%RSD) values) since the integration of smaller peaks yields a greater error on the fitted peak area estimation. The Frankie unit data shows the tendency to increased RSDs for Sn Sb, Ba and Pb. This might be due to the lower sensitivity of the spectrum after the peaks of the W anode (in the case of Ba L lines, it is their overlap with Ti K lines). The precision found in our set of data is normally higher than the instrumental precision obtained with other spectroscopic techniques (Table 1). Nevertheless, we can state here that the dispersion of the data is acceptable for accuracy considerations.

### 3.3. Accuracy with the Fundamental Parameters Approach

The FP method is a well-established quantification algorithm that can provide accurate quantitative results, especially when used in conjunction with matrix-matching standards [4,36]. Values calculated by PyMCA software were examined to assess their correspondence with the nominal ones. The result of this comparison is reported in Figure 4.

The 2σ error values for all the XRF units (95% confidence interval) are reported for each element. These values were calculated as two standard deviations of the absolute values of relative error for each data point. Many of the data points are within ±20% of the nominal values, but a significant number of them fall within ±50% of error. This quite broad range renders this data unacceptable for many applications. The accuracy values of each XRF unit differ for each element. It is remarkable that the Frankie unit values for the heaviest elements exceed the 100% limit. It should be kept in mind that Frankie unit values had lower precision for Sn, Sb, Ba, and Pb (Figure 3). The Unisantis unit is also demonstrating the broadest range of errors on many elements, and this might be connected to the lower number of acquisitions of the CMOG reference glasses (Frankie and Unisantis units) and the archaeological references (for Frankie unit). Due to the LOQ threshold implementation, Figure 4 lacks many data points. Several elements were not considered by some XRF units.

### 3.4. Ratio Coefficients Correction

It was decided to further look into the data in order to find the best way of improving the FP quantification of the datasets. By observing the data, it was noticed that some of the elements’ values were systematically under or overestimated within a single instrument’s dataset. This points out the limits of the FP model implemented through PyMCA: it accounts for a large number of variables, making the task of their optimization ever more laborious when approaching perfection. Instead, it was attempted to offset these variations by using the average ratios of nominal to acquired values of CMOG reference glasses to obtain the coefficients for normalization of all the values in the dataset. The formula for this type of correction is the following:$$x_{1}=x_{0}\frac{\left(\frac{A_{nom}}{A_{acq}}\right)+\left(\frac{B_{nom}}{B_{acq}}\right)+\left(\frac{D_{nom}}{D_{acq}}\right)}{3}$$
where $x_{1}$ is the corrected value of the element and $x_{0}$ is the initial value of the element. $A,\;B,D_{nom}$ and $A,\;B,D_{acq}$ are the nominal and acquired values of CMOG glasses, respectively. In this way, archaeological reference glasses were used as a validation set for the new accuracy check. Figure 5 provides the results of such a correction compared to the initial quantification. The values are compared as percentage recoveries from the nominal values. Four sets of recovery values were calculated for each XRF unit: two sets of initial quantification (average CMOG and ARCH recovery) and two corrected datasets (again, as an average of recoveries of two respective groups of samples).

The division of CMOG and archaeological reference groups was done to check the usefulness of such a correction. Since the coefficients were calculated on the CMOG set of three glasses, it was expected that the average recovery for the corrected values would be close to 100%. In such a case, one should look at the improvement of the data of the archaeological reference group. Indeed, corrected CMOG glasses’ recoveries for all instruments tend to align very near the 100% mark (red circles). As for the corrected archaeological reference glasses group, the resulting values mostly improve when compared to the values of initial quantification, but it is not always the case. For the Elio instrument, for example, an improvement in Cu recovery is “compensated” by a larger bias for Zn. However, most of the corrected values lie in the 20% interval from the nominal values (black circles) that was arbitrarily decided to be an acceptable deviation from the true values. Some corrected values are less than 10% off the nominal ones. Some elements do not have representation in the archaeological reference group of some XRF units (Co, Ni, Zr, and Ba were considered mostly only within the CMOG group). For such elements, it is impossible to assess the effect of correction. Hence, the data on these elements should not be considered as quantitative, yet the rest of the elements that are represented in the archaeological reference group do obtain better values after this type of correction, so it can be considered a useful transformation on the way to improving the overall accuracy.

### 3.5. Linear Regression Correction

Another type of empirical coefficient correction that was tested in this study is the application of linear regression coefficients. This approach was described elsewhere [58]. Here we will state only that we calculated slope and intercept values of the initial quantification and used an inverted linear regression equation to correct the data. The formula for this transformation is the following equation:$$x_{1}=x_{0}b_{0}+a_{0}$$
where $x_{1}$ is corrected value of the element, $x_{0}$ is the initial value of the element, and $a_{0}$ and $b_{0}$ are the intercept and slope values of the initial regression, respectively. Unlike the previous type of correction, this one is increasingly efficient when the number of points that constitute the calibration line grows. The elements that do not have at least two data points cannot undergo such transformation, starting from elements with three data points, it is possible to estimate the error of the quantification, which makes such data semi-quantitative. Among the benefits of this correction is the accountability of any residual matrix effect that was not accounted for during the FP quantification. Figure 6 shows the binary plots in which the nominal values are plotted with: the concentration determined as indicated in 3.3 (left row, initial quantification), the values obtained as indicated in 3.4 (middle row, ratio coefficients correction), and the values obtained with the regression coefficients described in this section (right row). The elements considered are K, Ca, Mn, Fe, Cu, Sr, and Pb. All of them are represented at least by five data points consistently in each instrument’s dataset. This makes it possible to build more reliable calibration curves. The nominal curve, which is created by plotting the nominal concentrations versus themselves in each plot, serves as the target value and creates a 20% deviation interval that was assigned to be acceptable. One can notice that initial quantification does not always provide the values that are within 20% of the deviation. In fact, the values can follow seemingly parallel curves (most pronounced in Sr initial plot) or diverge significantly from the nominal towards the upper point of the curve (K and Pb initial plots). Both situations reveal that the matrix effect was not fully accounted for. Both kinds of corrections tend to bring the points closer to the nominal curve while still featuring the outliers that eventually broaden the uncertainty margins to more than 20%. On most occasions, regression correction produced a better match with the nominal curve than the values calculated with ratio coefficients. It is expected that the best accuracy is achieved in the middle region of each calibration curve, while the likelihood of divergences is higher in the lower and higher margins.

In order to build the final dataset, we adopted a mixed approach: with less than four data points for one XRF unit, data were corrected by the application of ratio coefficients; when four (or more) data points were available, we used linear regression correction. Concentration values for elements that were obtained by the application of ratio coefficients lack adequate uncertainty values. Those elements are presented in concentrations approaching the LOQ, therefore high uncertainty is expected. Table 3 provides, relative to each element, its respective way of correction for each XRF unit as well as the uncertainty of measurement for every dataset corrected with the linear regression method. This table also includes Pearson correlation coefficients (r) calculated for the corrected XRF values and the nominal concentrations. In most of the cases, these coefficients indicated a very strong correlation between the variables (r = 0.9 and higher; some are approaching 1). In several cases, they are less than 0.9 but still highlight a strong correlation (e.g., Sr and Sb values for Frankie unit). This corresponds to a divergence of the respective trend line in Figure 6. The effect might be connected to a lower signal-to-noise ratio in the higher energy region of the spectrum produced by this unit.

It is worth noting that no systematic bias from the nominal values emerged for VA08 and VA07, which were analysed as chunks. Therefore, no significant bias determined by surface morphology or glass alteration emerged from the data.

### 3.6. Multivariate Approach to the Reproducibility Assessment

In order to check if the obtained reproducibility among the different types of equipment supports the purpose of investigating a large set of archaeological glass with different p-XRF units, we employed multivariate analyses. Principal Component Analyses (PCA) and Hierarchical Cluster Analyses (HCA) were used to check if the compositions of the same set of samples obtained from different p-XRF units will result in a proper representation (PCA) or classification (HCA) of the samples. In archaeometry, the compositions of glass are normally used to highlight similarities or differences among archaeological samples, therefore it is fundamental to assess if the use of different p-XRF units will prevent the possibility to draw any conclusions of archaeological relevance based on the dataset.

PCA and HCA are widely used tools in data exploration. PCA allows for reduction of dimensionality with a subsequent representation of all the data in a single space with two or three coordinates, whereas HCA organizes samples into groups based on how closely associated their compositions are and represents them in one single graph that highlights the relationships within the dataset [59,60].

In this particular instance, we employed both PCA and HCA to check if the data were from the same material, but different XRF units will be clustered together. Such group arrangements will support evidence of reproducibility for these datasets. On the other hand, distinguishing the reference samples will demonstrate the ability of quantitative p-XRF data to serve as a basis for archaeometric discussion on glass technology and, perhaps, even provenance. In order to perform the algorithm, the data corrected by linear regression or ratio coefficients (see Table 3 for the type of correction for each element/unit of equipment) were used, including the nominal values, that were used to check for any skewing. The elements included in the analysis were: K, Ca, Ti, Mn, Fe, Co, Ni, Cu, Zn, Sr, Sn, Sb, and Pb. Cells with no data were filled with “0”. The data were then scaled with a Standard Normal Variate (SNV) transformation.

Five PCs were created, with three bearing most of the explained variance (PC1: 52.65%, PC2: 20.62%, and PC3: 14.34%). The scores and loadings plots of PC1 against PC2 and PC3 are reported in Figure 7. There is a clear separation of sample groups instead of unit groups. Usually, groups are tightly packed without any visible skewing of a specific dataset. Exceptions are the VA samples that are clustered together, although VA70 can be distinguished using PC3. These glasses have a very similar composition and belong to the same type of (clear) glass. Therefore they cannot be differentiated based solely on the concentrations of the elements detected in this study [48,49,61]. CMOG B of Frankie unit can be separated from the rest of the CMOG B data points. The CMOG glasses, which represent different glass types, are properly located in different quarters of the PC1 vs. PC2 plot with respect to the point of origin based on their differences in composition.

Clusters were calculated using Euclidean distances on SNV pre-treated data and were based on at least three valid measures. It was decided to present eight clusters (by the number of reference materials). Figure 8 shows the result of the calculation. It can be seen that the datasets are grouped by sample. VA27 and VA08 belong to one cluster by using the same hierarchical level that usually divides single samples, but they are properly separated at a lower level of similarity. These reflect the subtle differences in composition as these samples are different only by Ti and Fe values (Table 1). The CMOG B data of the Frankie unit falls as an outlier between the CB36 cluster and the rest of the CMOG B acquisitions. However, in general, these cases do not impede datasets of the same sample, but different XRF units plus nominal compositions can be grouped together.

Both Figure 7 and Figure 8 show that the graphical representation of the data with multivariate methods is determined by the compositions of the samples, and no bias from the use of different equipment emerged. This allows for discussion of glass compositional groups (plant ash, natron, and potash) based on K and Ca values and (de)colorants and opacifiers (by the transition metal values), which is invaluable information about glass technology of the past [62].

## 4. Conclusions

The investigation highlights some limitations and pitfalls of the quantitative p-XRF method for archaeological glass analyses, which remains a challenging task not only in terms of quantification strategy for adaptation to a specific matrix in p-XRF analyses but also due to the intrinsic complexity of the interpretation of compositional data for ancient glass that results from the great variety of components in the glass batch, and therefore in a large quantitative variability.

On the other hand, by focusing on the analytical part, we demonstrated that, despite their hardware differences, four p-XRF units produced quite similar results provided that strictly controlled data handling is performed. Unfortunately, the set of certified reference materials available for elemental analyses of archaeological glass is very limited, and therefore it has not been possible to gather a large number of data points across all the elements of interest. Nevertheless, the results we presented in this work demonstrate that it is possible to achieve good-for-purpose reproducibility between different p-XRF units by following fairly practical procedures: ratio coefficients correction can be used on FP data when a low number of reference standards are available (less than four), whereas the data can be corrected by linear regression correction when four (or more) reference data points are available. This procedure, besides increasing the potential pool of quantitative compositional data on ancient glass, allows us to state that the compositions of glass determined through different p-XRF equipment can be actually used to highlight similarities or differences among archaeological samples (if the levels of uncertainty are lower than the actual differences between groups of glasses), and that the data obtained through different p-XRF units will allow us to draw conclusions of archaeological relevance based on the glass composition determined in situ.

## Figures and Tables

**Figure 1 molecules-27-06068-f001:**
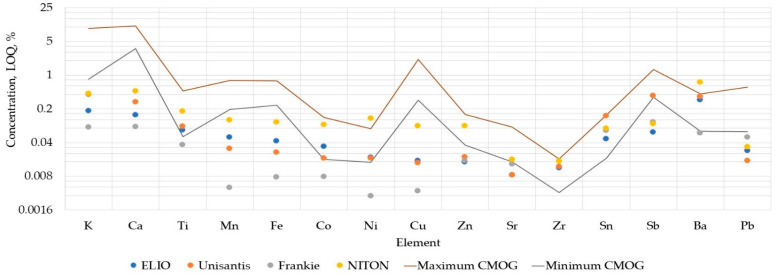
Comparison of the LOQ estimation of different units. The concentration range of CMOG A, B, and D reference glasses covers the space between the lines. Logarithmic scale.

**Figure 2 molecules-27-06068-f002:**
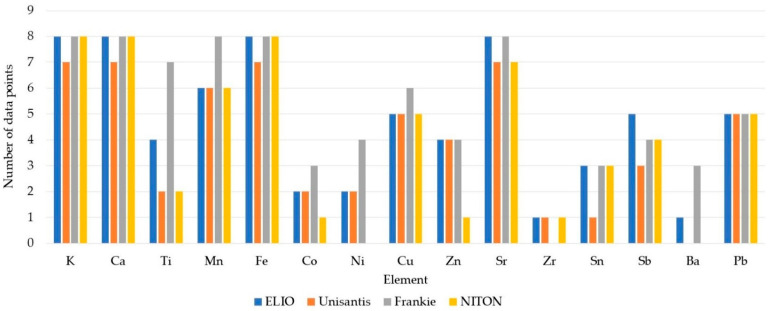
Number of data points considered in this study for each XRF unit with the division for each element of interest.

**Figure 3 molecules-27-06068-f003:**
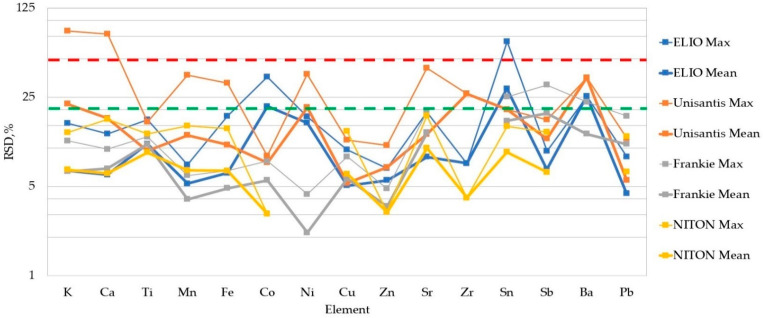
Estimation of the analytical precision for each XRF unit as the percentage difference from the mean value of repeated measurements. ELIO and NITON: ten replicates of CMOG reference glasses. The remaining data are calculated from the triplicate acquisitions. Frankie data only features CMOG values. The green dotted line is a 20% level of dispersion. The red dotted line is a 50% level of dispersion. Logarithmic scale.

**Figure 4 molecules-27-06068-f004:**
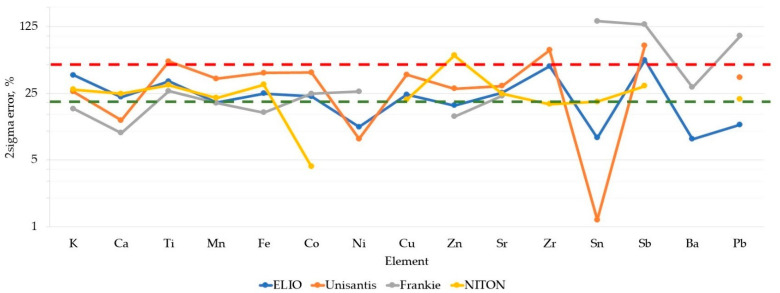
Accuracy assessment of the values of each XRF unit, calculated with the FP method. The green dotted line represents a 20% relative error (95% confidence). The red dotted line represents a 50% relative error (95% confidence).

**Figure 5 molecules-27-06068-f005:**
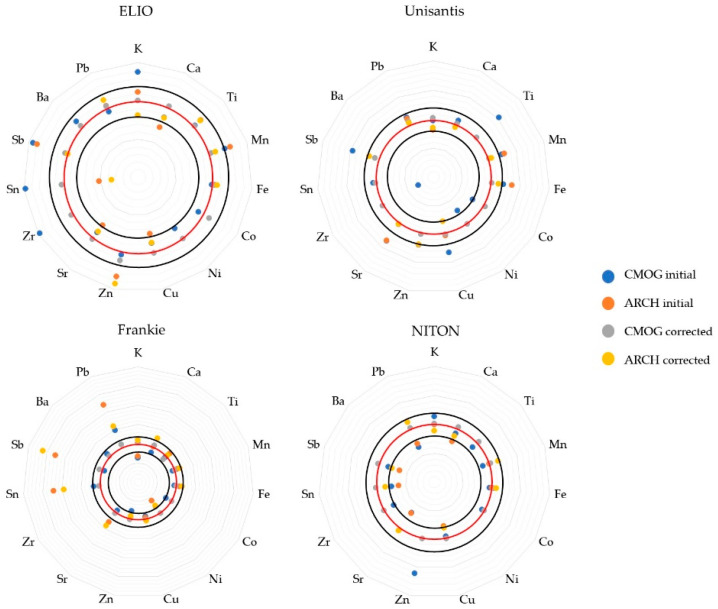
Spider plots demonstrate the recoveries of initial quantification compared to the values corrected by ratio coefficients. CMOG data points include averaged recoveries of CMOG reference glasses. ARCH data points include averaged recoveries from archaeological references. The red circles are the 100% recovery levels, while the inner and outer black ones are 80% and 120%, respectively.

**Figure 6 molecules-27-06068-f006:**
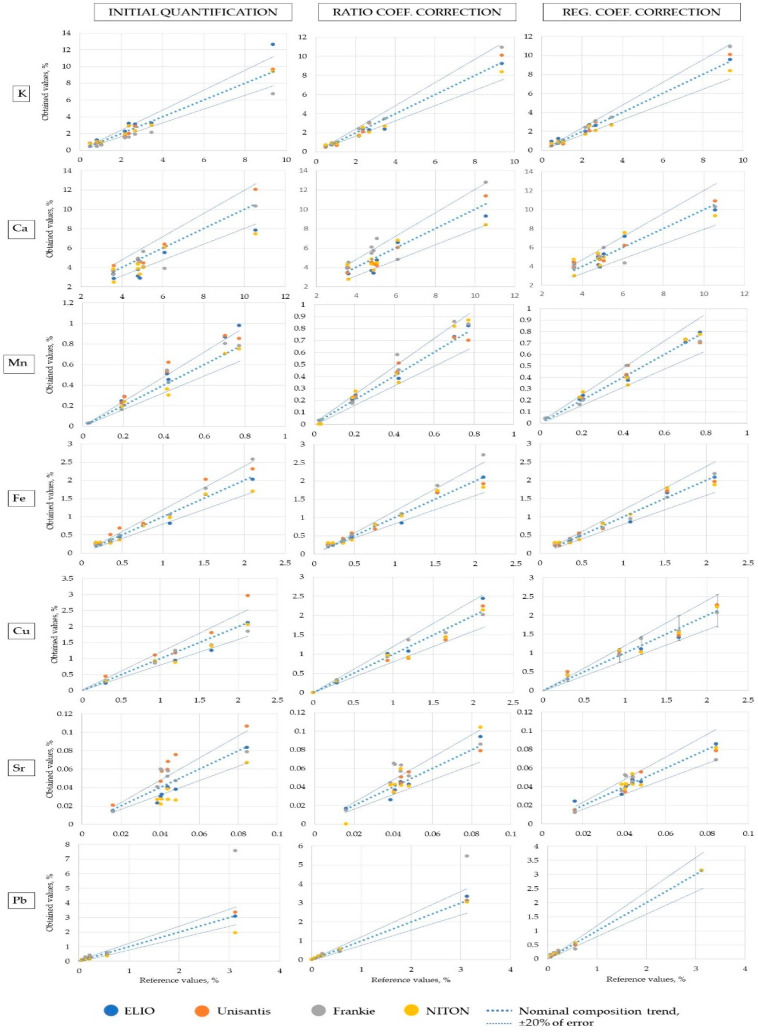
Binary plots of three kinds of quantification against the nominal concentrations. Nominal values are plotted against themselves to demonstrate the correct regression line. Twenty percent of the error along the nominal curve is marked by thinner dotted lines.

**Figure 7 molecules-27-06068-f007:**
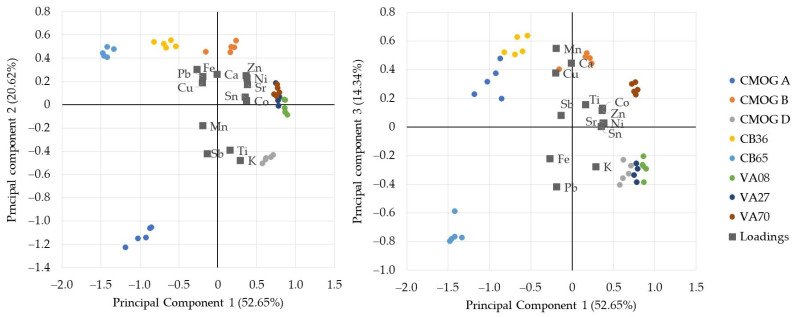
PCA of the dataset. Biplots for PC1 vs. PC2 (**left**) and PC1 vs. PC3 (**right**). Squares: loading. Circles: scores. Five points of each sample are from each unit of equipment (four) plus the nominal concentration (one). The numbers in parenthesis represent the variance percentage.

**Figure 8 molecules-27-06068-f008:**
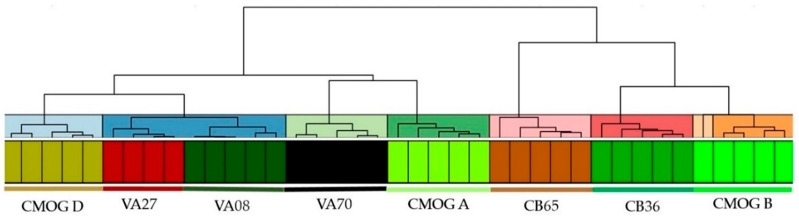
Hierarchical clustering plot that includes five datasets of each reference material (one for each XRF unit, except for VA27, which does not have values for Unisantis). The dendrogram features eight clusters distinguished by a different color (upper part). The bottom part shows color-coded datasets (each color represents a specific sample).

**Table 1 molecules-27-06068-t001:** Compositions (reported in element weight percent) of the reference glasses involved in this study (relevant elements). Compositions are from: CMOG A, B, and D—([47] and references therein), VA08, VA27—[48]; VA70—[49]; CB65—[50]; and CB36—[51]. The number of significant digits reflects analytical precision as reported in the listed publications.

RMs	K	Ca	Ti	Mn	Fe	Co	Ni	Cu	Zn	Sr	Zr	Sn	Sb	Ba	Pb
CMOG_A	2.38	3.60	0.47	0.77	0.76	0.134	0.02	0.94	0.035	0.09	0.004	0.150	1.32	0.41	0.067
CMOG_B	0.83	6.12	0.053	0.194	0.238	0.036	0.078	2.13	0.153	0.016	0.019	0.019	0.346	0.069	0.57
CMOG_D	9.4	10.6	0.228	0.43	0.36	0.018	0.039	0.304	0.080	0.048	0.009	0.079	0.73	0.261	0.224
VA_08	2.70	3.64	0.066	0.026	0.48	N.D.	0.002	N.D.	N.D.	0.044	0.005	N.D.	N.D.	0.009	N.D.
VA_27	3.49	4.81	0.114	0.036	1.09	0.001	0.006	N.D.	N.D.	0.039	0.010	0.001	N.D.	0.010	N.D.
VA_70	2.19	4.90	0.019	0.209	0.182	N.D.	0.001	0.004	N.D.	0.041	N.D.	N.D.	N.D.	0.009	0.003
CB_65	1.06	5.1	0.072	0.42	2.11	N.D.	0.003	1.20	0.007	0.041	0.006	0.110	0.51	0.021	3.13
CB_36	0.51	4.80	0.060	0.71	1.53	0.004	N.D.	1.66	0.040	0.044	0.006	N.D.	0.075	0.029	0.139

**Table 2 molecules-27-06068-t002:** Details of the XRF units involved in this study. V is the acceleration voltage of the X-ray source, and I is the intensity of the current.

ID	Hardware					Acquisition Parameters ^a^				
Device	Anode, V (max), I (max)	Beam Focusing, Focal Spot	Detector: Type, Active Area, Thickness	Resolution at Mn Kα	CPU Pulse Processing Channels	Spot Focusing Device(s)	Filter, Thickness	Time ^b^	V	I
ELIO	Rh 50 kV 50 µA	Pin hole 1.2 mm	SDD 25 mm^2^ 500 µm	140 eV	2048	Laser + Camera	none	90 s	40 keV	40 µA
Unisantis	Mo, 50 kV 1000 µA	Polycapillary 80 µm	Si-PIN 7 mm^2^ 300 µm;	186 eV	2048	Laser + Microscope	none	150 s	50 keV	300 µA
Frankie	W 50 kV 200 µA	Policapillary 300 µm	SDD 20 mm^2^ 450 µm	173 eV	4096	Laser	none	200 s	40 keV	80 µA
NITON	Ag 50 kV 100 µA	Pin hole 3–8 mm	SDD 25 mm^2^ 500 µm	185 eV	4096	Camera	Cu 125 µm	30 s	40 keV	50 µA
							Fe/Al 125 µm	30 s	20 keV	100 µA
							none	30 s	8 keV	100 µA

^a^ NITON acquires 4 spectra within a single acquisition, one of which is redundant for this study. ^b^ For the ELIO unit the time parameter is the total time of the acquisition (sum of live and dead time). For the rest of the units, the time parameter means live time.

**Table 3 molecules-27-06068-t003:** Corrections applied and the errors of the measurement (2σ values). L: linear regression coefficients. R: ratio coefficients. N/A: not applicable. Correlation coefficients are calculated between corrected and nominal values.

Element	ELIO			Unisantis			Frankie			NITON		
	Type of Correction	Pearson Correlation	Relative Error 2σ, %	Type of Correction	Pearson Correlation	Relative Error 2σ, %	Type of Correction	Pearson Correlation	Relative Error 2σ, %	Type of Correction	Pearson Correlation	Relative Error 2σ, %
K	L	0.990	54.5	L	0.996	27.4	L	0.998	34.0	L	0.993	35.7
Ca	L	0.946	15.8	L	0.987	13.2	L	0.932	19.7	L	0.905	19.5
Ti	L	0.994	32.1	R	0.983	N/A	L	0.995	24.3	R	0.983	N/A
Mn	L	0.993	12.4	L	0.978	15.5	L	0.988	36.7	L	0.974	23.7
Fe	L	0.986	33.6	L	0.989	6.5	L	0.994	27.4	L	0.977	33.6
Co	R	0.989	N/A	R	0.992	N/A	R	0.999	N/A	R	N/A	N/A
Ni	R	0.986	N/A	R	0.983	N/A	L	0.998	12.0	N/A	N/A	N/A
Cu	L	0.970	22.7	L	0.965	49.6	L	0.989	11.5	L	0.977	21.0
Zn	L	0.969	22.7	L	0.998	6.6	L	0.998	11.9	R	N/A	N/A
Sr	L	0.969	33.0	L	0.965	11.8	L	0.868	17.6	L	0.944	12.9
Zr	R	N/A	N/A	R	N/A	N/A	N/A	N/A	N/A	R	N/A	N/A
Sn	R	0.876	N/A	R	N/A	N/A	R	0.846	N/A	R	0.983	N/A
Sb	L	0.983	100.0	R	0.914	N/A	L	0.736	64.1	L	0.978	27.9
Ba	R	N/A	N/A	R	N/A	N/A	R	0.999	N/A	N/A	N/A	N/A
Pb	L	1.000	33.4	L	0.999	26.4	L	0.995	98.0	L	1.000	20.5

## Data Availability

Not applicable.

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
