# Peer review of "Data from Multiple Portable XRF Units and Their Significance for Ancient Glass Studies"

_molecules, 2022, doi:10.3390/molecules27186068_

Round 1

Reviewer 1 Report

A well structured manuscript is presented, with an in-depth comparison of analytical results obtained with 4 different XRF setups on glasses with known composition. The manuscript is sound and well in line with the aims and scope of the journal.

I recommend to accept it with minor revisions, provided that the comments below are addressed: 

line 51-58: Actually, this limitation is not so typical of portable systems (which are the topic of the paper), but is generally linked to any technique having to do with X-ray detection, that requires proper fitting

line 113-117: this, in fact, is the purpose statement of this study, i.e., "can we really compare data on archaeological collections, acquired with different instruments?". I would give it some more prominence in the text

line 142-144: Many glass studies rely on NIST 610 SRMs, in particular for trace analysis. Having a composition macroscopically not too far from the CMG B glass, it could have been a good standard for this study, allowing comparison of the results for many trace elements. Is there a specific reason why this has not been selected here?

Line 154-156: Why was CMG D used for this study? it is rather different, matrix-wise, from the archaeological glasses you measured (low Na, high K and Ca...)

Table 1: since these values will be used as real values for the evaluation of the accuracy of the 4 instruments, I think some information on the uncertainty of these values should be provided here, especially for the archaeological glasses that were previously measured with other techniques. 

Line 173: VA08 and VA27 were analyzed as chunks of glass, but the compositions in Table 1 are obtained with ICP-OES or ICP-MS (presumably bulk glass composition): besides surface shape, has the effect of surface alteration (depletion of Na, K...) been considered, causing a possible deviation from the tabulated composition? 

Section 2.2: what I am missing here is a bit of an introduction with the reason why these very 4 XRF instruments have been selected, besides being different technically. Are they representative of what the market can offer? Are they all commonly used for archaeometry studies?

line 207-210: why different number of replicates were acquired for different instruments? This can affect all the following comparisons, as you correctly point out in line 332-333

line 211-217: it's not clear what was done concerning the "missing" low Z elements. Was any of the possibilities mentioned at line 89-91 used, or were the nominal values used?

line 296 and following: how about the precision of the ICP-OES and ICP-MS measurements used for the nominal composition of the archaeological samples? It would be useful for the reader to have here a direct comparison with it, i.e., how much better it is than the XRF precision

line 346 and following: being CMG D more different matrix-wise from the archaeological glasses, I wonder if the same correction was attempted with CMG A and B only, and how the were the recoveries of the archaeological glasses, in comparison.

Conclusions: I would suggest recalling here more explicitly the purpose statement of line 113-117 and trying to answer it, i.e., is it possible to draw meaningful conclusions when comparing data acquired with different instruments? To me, the results seem encouraging, despite the several limitations. Some suggestions concerning instrumental and acquisition conditions could also be offered in the conclusions.

Reviewer 2 Report

Dear Editors,

The paper submitted, although not innovative in terms of a research topic, being more technical in its nature, is referred for publication on a special issue entitled Chemical Analysis Strategies in the Cultural Heritage Field, maybe finding a justification there.

In my opinion it deals essentially with the comparison of results obtained with different X-ray fluorescence equipment the research group has access, trying to establish different methodologies for chemical analysis of ancient glass.  Regarding the proposed approach, it is scientifically sound and well reported. The text is well written, and I find no fault in its presentation.

If the Editors find it useful for the special issue in question, although the paper seems astray from the usual publications in this journal, I do not object to its publication in its present form in this special issue.

Reviewer 3 Report

The authors have done a sirgnificant amount of work. However, there are now  new results about archaeological glasses. The main part of the manuscript concerns that statistical analysis of the data. However, no new statistical methods are introduced. It is essntially a demonstration of best practices for pXRF.
